# Feasibility and Outcomes of the Early Start Denver Model Delivered within the Public Health System of the Friuli Venezia Giulia Italian Region

**DOI:** 10.3390/brainsci11091191

**Published:** 2021-09-10

**Authors:** Raffaella Devescovi, Vissia Colonna, Andrea Dissegna, Giulia Bresciani, Marco Carrozzi, Costanza Colombi

**Affiliations:** 1Division of Child Neurology and Psychiatry, Institute for Maternal and Child Health—IRCCS “Burlo Garofolo”, 34137 Trieste, Italy; vissia.colonna@burlo.trieste.it (V.C.); giulia.bresciani@burlo.trieste.it (G.B.); marco.carrozzi@burlo.trieste.it (M.C.); ccolombi@med.umich.edu (C.C.); 2Department of Life Sciences, University of Trieste, 34127 Trieste, Italy; andrea.dissegna@phd.units.it; 3IRCCS Fondazione Stella Maris, Calambrone, 56128 Pisa, Italy; 4Department of Psychiatry, University of Michigan, Ann Arbor, MI 48109, USA

**Keywords:** autism spectrum disorder, early intervention, Early Start Denver Model, preschoolers, public health system, ESDM Curriculum Checklist

## Abstract

The Early Start Denver Model (ESDM) is an evidence-based early intervention model for young children with autism spectrum disorder (ASD). It is crucial to investigate the feasibility of the ESDM in community settings in contexts that are culturally different from American universities in which the model was originally developed. The aim was to further evaluate the effectiveness of the ESDM delivered within the Italian community setting at low intensity. We compared a group aged 19 to 43 months receiving the ESDM for 2 h per week over the course of 1 year with a concurrent, comparable, non-randomized control group receiving treatment as usual (TAU). Children were evaluated at baseline (T0) and after 6 months (T1) and 12 months (T2) of intervention. Feasibility was evaluated by parent and therapist questionnaires, retention rate, and therapist treatment fidelity. Both groups made similar gains in cognition and language abilities. The ESDM group made larger improvement in domains measured by the ESDM Curriculum Checklist, including communication, social skills, and maladaptive behaviors. Feasibility seemed well supported by retentions, therapists and parent satisfaction, and treatment fidelity. Our study further supports the feasibility of the ESDM implemented within the Italian public health system and suggests a better response in the ESDM-treated group than in the control group.

## 1. Introduction

Autism spectrum disorder (ASD) is a heterogeneous neurodevelopmental disorder characterized by impairments in social interaction and communication, as well as by the presence of repetitive patterns of interests and behaviors [1]. The number of children diagnosed with ASD has steadily increased over the last two decades, with an estimated prevalence of 1 out of 54 individuals in the USA [2] and 1 out of 77 in Italy [3]. ASD is a lifelong condition bringing elevated financial costs to society and important burden to families. Despite rigorous research studies having shown that specialized early intervention that begins within the first three years of life positively impacts developmental trajectories [4,5], on average children still access services at approximately 5 years of age worldwide, and often such services are not specific for ASD [6].

Most research evidence supporting the impact of early intervention for children with ASD comes from university-based efficacy studies performed with much higher resources in comparison to those usually available in communities [5]. Currently, there is a large gap between findings of efficacy studies and their translation into community settings. The first step to reduce such gap is to demonstrate, through effectiveness studies, that interventions developed and evaluated within universities can be implemented within community settings and can lead to positive outcomes for children with ASD and their families.

Over the last few years, a limited number of studies has shown that it is possible to effectively implement evidence-based early intervention models within community settings. Examples of models evaluated in community settings are the Pivotal Response Training (PRT) [7] and the Joint Attention, Symbolic Play, Engagement and Regulation intervention (JASPER) [8].

In Italy, the Early Start Denver Model (ESDM) has been implemented and evaluated in three different community studies. Devescovi et al. [4] found that a group of 21 children with ASD, aged 20–36 months, who received the ESDM intervention for 3 h per week over 15 months, made significant improvements in language and cognitive functioning. Moreover, the authors reported that the youngest children and those with lower cognitive functioning at baseline assessment were those who showed more significant changes in comparison to the older children and those with less significant impairments, respectively, in the sample. Colombi et al. [9] evaluated the outcomes of 22 young children with ASD receiving ESDM in a center-based context for 6 h per week over 6 months. The ESDM group was compared to a group of 70 young children diagnosed with ASD who received treatment as usual with similar intensity over 6 months. After 3 months and 6 months of treatment, children in both groups improved in cognitive, adaptive, and social skills. Children in the ESDM group made larger gains in cognitive, language, and social skills after 3 months and 6 months of treatment. Moreover, the ESDM group made larger gains in adaptive skills after 3 months of treatment. Finally, Contaldo et al. [10] examined outcomes in 32 children diagnosed with ASD after one year of community-based ESDM intervention and identified several predictors of treatment response including pre-treatment non-verbal abilities, symptom severity, action and gesture repertoire, and lexical comprehension.

These studies suggest that Italian health professionals can learn the ESDM and that this evidence-based specialized intervention can be implemented within Italian communities and lead to better outcomes in comparison to treatment as usual (TAU). In most areas of Italy, children with ASD receive low-intensive, nonspecialized treatment. In general, the Italian Public Health System offers between 2 and 6 h of intervention composed of speech therapy, educational services, and neuropsychomotricity.

The ESDM is an empirically based, manualized play- and daily activities-based intervention that fuses developmental and relationship-based models with principles and practices of applied behavior analysis (ABA). The ESDM is delivered by adults within the context of play and daily routines in which highly precise naturalistic behavioral teaching is imbedded, making this one of the evidence-based naturalistic developmental behavioral interventions (NDBI) [11]. The ESDM has been implemented in a variety of settings, including intensive autism-specialty delivery [12], daycare centers and preschools [13], parent education [14,15] and telehealth [16]. Moreover, the ESDM can be implemented by a variety of professionals such as psychologists, occupational therapists, early childhood educators, behavioral analysts, and speech therapists [17]. The meta-analysis conducted by Fuller et al. [18] examines the effects of ESDM on young children with ASD implemented in a variety of delivery settings on a variety of outcome measures. The authors find moderate and significant improvement for ESDM-treated children compared to control groups, especially in cognition and language, while no significant changes are found in autistic symptomatology, adaptive behavior, social communication, and repetitive and restricted interests (RRB).

Here, we report the results of S.F.I.D.A. (Screening, Friuli Venezia Giulia, Intervention, Diagnosis, Autism), an early detection and intervention program conducted in the public health system of Friuli Venezia Giulia (FVG), a region located in the northeast area of Italy. S.F.I.D.A. extends our previous work [4] by offering ESDM training to the whole FVG region, further evaluating the feasibility of the intervention delivered within the public health system and evaluating treatment outcomes produced by such intervention in comparison to TAU administered at a similar intensity. Originally, the S.F.I.D.A. program aimed to recruit 100 children. However, due to the disruption of COVID-19, here we report results of a smaller sample of children who completed the intervention before the upsurge of the health emergency in March 2020.

The primary goal of our study was to further evaluate the feasibility of the ESDM within the Italian Public Health System. Feasibility was operationalized on the basis of guidelines outlined in Bowen et al. [19] and those used in Vivanti et al. [5] and Colombi et al. [9]. We utilized the following variables: (1) acceptability (how the intervention is accepted by the individuals involved in the program), (2) demand (to what extent is the program likely to be chosen), (3) implementation (the degree of execution of the intervention in terms of the manualized procedures), (4) practicality (the extent to which delivery of the program can be implemented in a specific context), (5) adaptation and integration (the measure to which the intervention can be integrated within the existing system). The feasibility was assessed through a combination of surveys, external evaluation reports, and analysis of services utilization.

The secondary goal of our study was to evaluate the effectiveness of ESDM treatment implemented at low intensity over the course of 1 year in a large Italian community by comparing the outcomes of the children who received the ESDM with those of children who received TAU. Our study further attempts to evaluate the ESDM effectiveness delivered within the Italian Public Health System to young children with ASD at very low intensity, two hours per week, mapping the amount of treatment currently available in the Friuli Venezia Giulia Region. On the basis of previous research, we hypothesized that children in the ESDM group would demonstrate higher gains in language, cognitive, social, and daily living skills.

## 2. Materials and Methods

### 2.1. Design

This is a quasi-experimental controlled treatment study of the ESDM delivered within the public health system of the Friuli Venezia Giulia (FVG) Italian region. The study received funds for clinical, translational, basic, epidemiological, and organizational research on the basis of the Regional Law 17/2014 of the FVG region, and it was approved by the Single Regional Ethics Committee (CEUR-2017-PR-050-BURLO).

The original design involved recruiting all children who qualified for the study in the Friuli Venezia Giulia region between March 2018 and December 2019 and monitoring them for 12 months over the course of the intervention. The experimental group received the intervention in regional centers that had received ESDM training prior the beginning of the study. The control group received services in regional centers that were waiting to receive ESDM training and delivered TAU at the time of the study.

During this period, 85 children, aged 19 to 43 months, were enrolled in the study after obtaining parental written consent. The ESDM group included 37 children, while the TAU 48. However, 7 out of 85 families dropped off the study. In the ESDM group 3 families withdrew their consent to analyze their data despite continuing the intervention. In the TAU group, 2 families preferred to interrupt intervention to access private services at higher intensity, while 2 additional families moved their residence outside the FVG region. We expected to complete the 12-month follow-up for the entire sample by December 2020, but since the beginning of the pandemic lockdown imposed by the Italian government in late February 2020, it was not possible to provide in person treatment continuously, nor to assess children at specific time points in the public services. Prior the beginning of the lockdown, 19 children in the ESDM group and 35 in the TAU had already completed the 12-month period of treatment and the last follow-up assessment.

We realized that the two groups of children who completed the study were unbalanced in number and age. In particular, children in the TAU group were older than children in the ESDM group (TAU: mean = 34.22 months, SD = 5.06 months; ESDM: mean = 29.68 months, SD = 6.53 months; *p* = 0.013). Therefore, we randomly extracted a subsample of 19 children from the TAU group that matched the age of the ESDM children. We assigned a progressive number to each child in the TAU group and generated several lists of 19 random numbers from this progression. We selected the first list of children with an average age within 1 SD of the average age of the ESDM group. Therefore, the final sample included 38 children equally distributed in two groups matched by age (19 ESDM vs. 19 TAU).

We enrolled children who met ASD diagnostic criteria according to the DSM-5 [1] and met cut-off for ASD on the Autism Diagnostic Observation Schedule—Second Edition (ADOS-2) [20,21] administered by experienced clinicians prior the beginning of the study. The diagnostic evaluation protocol, consisting of all the tools listed below among the measures of effectiveness, was administered at baseline time (T0) before starting treatment, and it was repeated after 6 months (T1) and after 12 months (T2) of treatment. In both the ESDM and TAU groups, children received treatment delivered by therapists for 2 h per week. Each treatment team included psychologists, speech therapists, and psychomotricity therapists. None of the recruited children had received any type of treatment before entering the study.

### 2.2. Participants

Inclusion criteria for children enrolled in the study for both groups were (a) diagnosis of ASD made by qualified professionals using standard valid and reliable assessment tools, (b) parent’s agreement to participate in the intervention for 12 months, (c) hearing and vision screened within the normal range, and (d) ability to use hands and ambulate. Exclusion criteria included (a) known neurological disorders (e.g., epilepsy), and (b) significant sensory or motor impairment (e.g., cerebral palsy).

The ESDM group comprised 19 children (17 males) between the ages of 19 and 43 months who had received an autism spectrum disorder diagnosis. The TAU group comprised 19 children (15 males) between the age of 24 and 36 months with similar inclusion and exclusion criteria of those in the ESDM group as outlined above. The socio-demographic characteristics of the sample are described in Table 1.

### 2.3. Measures

#### 2.3.1. Feasibility

Feasibility was documented through analysis of service utilization and external evaluation of the program, as well as through questionnaires based on Holzinger et al. [22] administered to parents and therapists (Table 2).

As *acceptability* indicators, we adopted the retention rate of the children enrolled and questionnaire results from parents and staff members at post-treatment. *Implementation* parameters were documented through fidelity ratings by the ESDM international trainer. *Practicality* of the program was intended in terms of the maintaining of the principles and strategies of ESDM and of how the intervention is compatible with the daily family activities and the staff’s regulatory restrictions; data were derived from the same questionnaires above. The *demand* indicator was the applicability among the potential users (i.e., children with ASD living in Trieste and Gorizia, the cities in which the ESDM was implemented). *Adaptation and integration* referred to how much the system needed to change to integrate the ESDM procedures into the pre-existing early intervention protocols; changes were documented internally in these protocols. *Efficacy* as perceived by parents and therapists was measured by the same questionnaires above.

#### 2.3.2. Effectiveness


*Bayley Scales of Infant and Toddler Development—Third Edition (Bayley-III)*


The Bayley-III [23,24] is a standardized instrument that measures early cognitive development in children from 16 days to 42 months of age. We calculated composite scores in the cognitive and language areas.


*Wechsler Preschool and Primary Scales of Intelligence—Third Edition (WPPSI-III)*


We used the WPPSI-III [25,26] to measure cognitive development in children 42 months of age and older. The WPPSI-III provides verbal, performance, and full-scale IQ scores. Seeing as the translated and validated version was available from 2019, the authors decided to continue the study with the administration of WPPSI-III. Depending on the child’s developmental stage, we adopted the Bayley-III or the WPPSI-III to assess cognition and language, as proposed by Kitzerow et al. [27].


*Autism Diagnostic Observation Schedule—Second Edition (ADOS-2)*


The ADOS-2 [20,21] is a standardized diagnostic observational instrument that measures ASD symptoms communication, social reciprocity, play, and restricted behavior. We compared ASD severity across participants by using the ADOS calibrated severity algorithm [28], which allows for comparison across different ADOS modules. We used the Calibrated Severity Scores Overall (CSS Overall), as well as the CSS for the Social Affect algorithm (CSS SA) and CSS for the Restricted and Repetitive Behavior algorithm (CSS RRB) calculated on the basis of procedures described by Hus et al. [29] and Esler et al. [30].


*Vineland Adaptive Behavior Scales—Second Edition (VABS-II)*


The VABS-II [31,32] is a structured standardized parent interview measuring adaptive behavior from birth to adulthood. The 11 subscales of the VABS-II are clustered into four domains: communication, daily living, socialization, and motor skills. The measure provides standard scores, age equivalents, and an adaptive behavior composite (ABC).


*ESDM Curriculum Checklist*


The ESDM Curriculum Checklist [33] is composed of 446 items that investigate the following domains: receptive and expressive communication, joint attention, social skills, imitation, cognition, play, fine and gross motor skills, behavior, and personal independence. The ESDM Curriculum Checklist is a developmental scale that evaluate children in all areas of development with skills organized through a developmental sequence corresponding to abilities that typically developing children demonstrate between 12 and 48 months of age. For each item, we assigned a score of 2 to those items that referred to skills learned and generalized across people and settings (previously coded “+”), a score of 1 to those items that referred to emerging skills (previously coded “+/−”), and a score of 0 to the items that referred to absent skills. On the basis of procedures published by Contaldo et al. [10], we obtained five scores: communication (C), which includes receptive and expressive communication areas; socialization (S), which includes imitation, joint attention, and social areas; cognition and play (CP), which includes play and cognitive areas; motor (M), which includes fine and gross motor areas; and a total score (TS), which is the sum of the four previous domains. For the experimental group, the patterns of strengths and weaknesses identified by the ESDM Checklist were used to develop individualized treatment of objectives for each child. The checklist was administered by the therapists to both groups to further evaluate the developmental level of each child at T0, T1, and T2.


*Behavior Observation of Social Communication Change (BOSCC)*


The BOSCC [34] is a treatment response measure developed to measure changes in social communication behaviors in minimally verbal children. We received permission from Western Psychological Services and Dr. Catherine Lord to use a preliminary version of the measure (August 2016). The BOSCC was developed by modifying and expanding codes from the ADOS-2 [20]. Items are coded on a 6-point scale from 0 to 5, with higher scores reflecting more atypical behavior. It consists in 12 items that provide a total score, as well as scores in social communication (SC) and restricted and repetitive behaviors (RRB). We coded the BOSCC from 15 min videos of parent–child interaction using the same toys across dyads and times. Of the 15 min videos, we coded the 10 min obtained by eliminating the first and last 2.5 minutes. The scores of the two 5 min segments of each interaction were summed and averaged to obtain the total score, as well as the SC and RRB scores. All interactions between the child and caregiver were recorded with the same caregiver (mother or father). Training was led by one of the original developers of the measure who participated in BOSCC training at the Center for Autism and the Developing Brain (CADB). For the current study, three research assistants with master’s level education coded the videos after establishing reliability on the basis of the authors’ criteria: (1) scores within 1 point for more than 80% of the items on a given segment, and (2) total scores within 3 points. Each coder had to meet both criteria on six consecutives 5 min segments.

### 2.4. Intervention

*ESDM*. Each child participating in the ESDM group received 2 one-hour sessions per week of ESDM treatment delivered individually in a center-based context for 12 months. The treatment team included fully credentialed professionals with backgrounds in behavioral principles, clinical psychology, and developmental psychology. Prior to implementing the intervention, all therapists had first received formal ESDM training and had reached fidelity. The last author (C.C.), a certified ESDM trainer, provided the initial training and trained the team to fidelity of implementation according to procedures described in Dawson and Rogers [35].

A qualified therapist administered the ESDM curriculum assessment every 12 weeks. On the basis of the results obtained, we developed individualized learning objectives for each child and monitored them during each therapy session. The learning objectives focused on verbal and nonverbal communication, joint attention, social engagement, imitation, play, cognition, motor development, and adaptive skills.

According to procedures described by Rogers and Dawson [17], the therapists delivered the ESDM within the context of play and daily routines in which highly precise teaching is embedded. The therapists shared treatment objectives with the parents and discussed ESDM strategies with them at the end of each session. Parents were not formally trained in delivering ESDM, but they observed the therapy. Additionally, ESDM objectives and strategies were shared with the children’s teachers.

*TAU*. The TAU consisted of community-based interventions delivered by child neuropsychiatric services. The TAU group children, similarly to the ESDM group, participated in intervention for 2 one-hour sessions per week. The TAU teams were similar to the ESDM group in terms of professional background, except for the ESDM training. The TAU group is representative of the services usually delivered in Italy, where licensed professionals administer nonspecialized ASD interventions consisting in speech therapy, educational interventions, and psychomotricity. Psychomotricians are licensed professionals in Italy and in other European countries. Each child’s program includes individual goals and treatment objectives but is mainly based on staff expertise rather than manualized treatment strategies.

### 2.5. Statistical Analysis

Data from the clinical outcomes were first screened for significant skewness and kurtosis. As no violation of normality assumptions was identified, a mixed ANOVA with one within-subject factor (Time, with 3 levels: T0, T1, T2) and one between-subject factor (Group, with 2 levels: ESDM and TAU) was used to model the average change in the score of participants over time. A series of post-hoc Tukey’s tests were performed to further analyze the Group x Time interaction when multiple comparisons were involved.

Because of the COVID-19 pandemic, our final sample size was downsized from what was initially estimated. The reduction of the statistical power of our study made our analyses prone to type 2 probability error—that is, failure to reject a false null hypothesis. Thus, for a limited set of variables (CSS SA, CSS RRB, and VABS II—Daily living skills) in which the treatment effect was significant, but the interaction was not, we performed planned orthogonal *t*-tests to detect group differences at specific time points [36,37]. We provided effect sizes (η_p_^2^ and Cohen’s d) for each statistical model to facilitate future meta-analyses. Data analysis was performed in JASP v. 0.13.1 (https://jasp-stats.org/ accessed on 16 May 2021).

## 3. Results

### 3.1. Feasibility

All the parent and therapist questionnaires’ results are reported in Table 3.

*Acceptability.* The retention rate of 19 children in the ESDM group who completed one-year follow up was 100%. Parent and therapist questionnaires revealed mean score of 3.71 for the parents and 3.70 for the therapist, respectively, on a 4-point Likert scale, with score of 4 indicating “strongly agree with the statement”. Parents considered the ESDM as an appropriate approach for their children and appreciated to be actively involved in the treatment. The same positive judgement was expressed by the staff members.

*Implementation.* The ESDM therapists had already achieved certification prior to implementing the study. Fidelity was monitored during the implementation by the last author, C.C., an ESDM trainer, independent of treatment delivery.

*Practicality*. This was expressed in terms of how ESDM goals were embedded into daily family routines (mean score for parents: 3.88) and into therapist practices (mean score: 2.85). While parents overall reported high level of appreciation of the ESDM program, some parents shared that it was difficult to implement the strategies at home. Despite therapists reporting limitations in staff resources and sometimes in availability of play space and materials, they still believed that the children were making more progress as a result of the ESDM treatment (efficacy mean score: 3.66).

*Demand*. Since the beginning of the implementation of ESDM in public services in the Trieste and Gorizia areas in 2012, more than 200 families have received this intervention model. Given that the demand for treatment far exceeds the ability of services to provide it, at the moment it is only possible to administer the ESDM at low intensity.

*Adaptation and Integration*. Due to limitation in the number of staff members available, weekly treatment intensity was much lower than is recommended in the literature for early intervention in ASD. However, therapists still felt comfortable implementing the ESDM principles and strategies, and no major adjustments were needed. The integration of the ESDM into the working practices of public services for early intervention in autism is evidenced by the SFIDA project, which is an extension of the pilot project developed a few years earlier in the Trieste and Gorizia cities.

### 3.2. Effectiveness

We report all the results in Appendix A, which can be found in the Appendix A.

#### 3.2.1. Cognition and Language

*Cognition:* Both groups of children improved their cognitive score over time, as attested by a significant main effect of the factor Time (*F* (2, 72) = 4.58, *p* = 0.013, η_p_^2^ = −0.11). There was no significant difference between groups (Group: *p* = 0.957) and the Group x Time interaction was not significant (*p* = 0.106).

*Language*: The language score of both groups improved over time (Time: *F* (2, 72) = 17.81, *p* < 0.001, η_p_^2^= 0.34), but there was no significant difference between groups (Group: *p* = 0.442), and the Group x Time interaction was not significant (*p* = 0.769).

#### 3.2.2. ADOS-2 Calibrated Severity Score (CSS)

*CSS SA Affect*: There was a significant main effect of the factors Time (*F* (2, 72) = 30.36, *p* < 0.001, η_p_^2^ = 0.46) and Group (*F* (1, 36) = 4.19, *p* = 0.048, η_p_^2^= 0.10). The Group x Time interaction was not significant (*p* = 0.173). A set of a priori contrasts revealed that the two groups of children had the same baseline performance in T0 (*t* (36) =0.61, *p* = 0.550, *d* = 0.20), but not in T1 and T2. In particular, the average CSS SA score was lower for the ESDM group than the TAU group (overall mean difference = −1.53, *t* (45.87) = 2.58, *p* = 0.013).

*CSS RRB*: There was a significant main effect of the factor Group (*F* (1, 36) = 4.88, *p* = 0.034, η_p_^2^ = 0.12). However, both the factor Time (*p* = 0.09) and the Group × Time interaction (*p* = 0.49) were not significant. A set of a priori contrasts revealed that although the two groups of children had the same baseline performance in T0 (*t*(36) =−1.74, *p* = 0.090, *d* = −0.57), the overall CSS RRB score for the ESDM group was higher than the TAU group score in T1 and T2 (overall mean difference = 1.24, *t*(44.71) =−4.94, *p* = 0.041).

*CSS TOT*: Both groups improved their score over Time (*F* (2, 72) = 19.57, *p* < 0.001, η_p_^2^ = 0.35), but neither the factor Group (*p* = 0.378) nor the Group × Time interaction (*p* = 0.560) were significant.

#### 3.2.3. VABS-II

*Communication:* The communication score of both groups improved over Time (*F* (2, 72) = 4.22, *p* = 0.019, η_p_^2^ = 0.11), while the factor Group (*p* = 0.221) and the Group × Time interaction were not significant (*p* = 0.767).

*Daily living skills:* There was a significant main effect of the factor Group (*F* (1, 34) = 7.29, *p* = 0.011, η_p_^2^ = 0.18). On the contrary, the factor Time (*p* = 0.06) and the Group x Time interaction (*p* = 0.93) were not significant. Children in the TAU group obtained on average higher scores in T0 (*t* (34) = 3.34, *p* = 0.002, d = 1.15) and T1 (*t*(36) = 2.12, *p* = 0.041, *d* = 0.69) but not in T2 (*t*(36) =2.00, *p* = 0.052, *d* = 0.65), attesting that the ESDM and TAU group reached the same score at the end of the treatment.

*Socialization:* The socialization score of both groups did not improve over time (*p* = 0.385). Similarly, there was no significant difference between groups (Group: *p* = 0.252), and the Group × Time interaction was not significant (*p* = 0.696).

*Adaptive behavior composite*: Children of both groups improved their score over time (*F* (2, 72) = 3.89, *p* = 0.025, η_p_^2^ = 0.10), but there was no significant difference between groups (Group: *p* = 0.087), and the Group × Time interaction was not significant (*p* = 0.296) as well.

#### 3.2.4. ESDM Curriculum Checklist

*Receptive communication area:* There was a significant main effect of the factor Time (*F* (2, 66) = 68.31, *p* < 0.001, η_p_^2^ = 0.67) and the Group x Time interaction (*F*(2, 66) = 3.67, *p* = 0.031, η_p_^2^ = 0.10). A post hoc analysis revealed that the ESDM group significantly improved from T0 to T2 (T0 vs. T1: mean difference = −21.37, *p_t_* < 0.001; T1 vs. T2: mean difference = −12.94, *p_t_* < 0.001), whereas for the TAU group, the average score changed from T0 to T1 (mean difference = −12.12, *p_t_* = 0.013) but not from T1 to T2 (mean difference = −9.56, *p*_t_ = 0.091).

*Expressive communication area:* There was a significant effect of the factor Time (*F* (2, 66) = 47.53, *p* < 0.001, η_p_^2^ = 0.59) and the Group x Time interaction (*F*(2, 66) = 5.60, *p* = 0.006, η_p_^2^ = 0.14). The improvement from T0 to T2 was significant for the ESDM group (T0 vs. T1: mean difference = −18.53, *p_t_* < 0.001; T1 vs. T2: mean difference = −16.31, *p_t_* < 0.001) but not for the TAU group (T0 vs. T1: mean difference = −8.44, *p_t_* = 0.275; T1 vs. T2: mean difference = −8.62, *p_t_* = 0.253).

*Joint attention area*: Children of both groups improved their score (*F* (2, 66) = 40.71, *p* < 0.001, η_p_^2^ = 0.55). There was no significant difference between groups (Group: *p* = 0.646), and the Group x Time interaction was not significant (*p* = 0.484).

*Social skills area*: There was a significant effect of the factor Time (*F* (2, 66) = 34.71, *p* < 0.001, η_p_^2^ = 0.51) and the Group x Time interaction (*F* (2, 66) = 3.16, *p* = 0.049, η_p_^2^ = 0.09). The improvement from T0 to T2 was significant for the ESDM group (T0 vs. T1: mean difference = −19.39, *p_t_* < 0.001; T1 vs. T2: mean difference = −11.63, *p_t_* = 0.047) but not for the TAU group (T0 vs. T1: mean difference = −8.06, *p_t_* = 0.423; T1 vs. T2: mean difference = −9.25, *p_t_* = 0.271).

*Imitation skills area*: There was a significant effect of the factor Time (*F* (2, 66) = 33.03, *p* < 0.001, η_p_^2^ = 0.50) and of the Group x Time interaction (*F* (2, 66) = 3.40, *p* = 0.039, η_p_^2^= 0.09). The improvement from T0 to T2 was significant for the ESDM group (T0 vs. T1: mean difference = −6.26, *p_t_* = 0.002; T1 vs. T2: mean difference = −5.84, *p_t_* = 0.004) but not for the TAU group (T0 vs. T1: mean difference = −1.81, *p_t_* = 0.889; T1 vs. T2: mean difference = −4.62, *p_t_* = 0.080).

*Cognitive area*: Children of both groups improved their score over time (*F* (2, 66) = 37.56, *p* < 0.001, η_p_^2^ = 0.53). There was no significant difference between groups (Group: *p* = 0.157), and the Group x Time interaction was not significant (*p* = 0.278).

*Play area*: Both groups improved their score over time (*F* (2, 66) = 52.07, *p* < 0.001, η_p_^2^ = 0.61). There was no significant difference between groups (Group: *p* = 0.108), and the Group x Time interaction was not significant (*p* = 0.110).

*Fine motor skills area*: Both groups improved their score over time (*F* (2, 66) = 58.65, *p* < 0.001, η_p_^2^ = 0.61). There was no significant difference between groups (Group: *p* = 0.432), and the Group x Time interaction was not significant (*p* = 0.598).

*Gross motor skills area*: Children improved their score over time (*F* (2, 66) = 30.90, *p* < 0.001, η_p_^2^ = 0.48). Neither the Group factor (*p* = 0.962) nor the Group x Time interaction (*p* = 0.130) were significant.

*Behavior skills area*: There was a significant main effect of both Time (*F* (2, 66) = 10.41, *p* < 0.001, η_p_^2^ = 0.24) and Group (*F* (2, 33) = 5.11, *p* = 0.031, η_p_^2^ = 0.13). While the two groups had a similar baseline score in T0 (*t* (34) = −0.95, *p* = 0.351, *d* = −0.32), the ESMD group scored higher than the TAU group in T1 and T2 (*t* (44.61) = −2.58, *p* = 0.013).

*Personal independence area*: Children improved their score over time (*F* (2, 66) = 22.66, *p* < 0.001, η_p_^2^ = 0.41). Neither the Group factor (*p* = 0.264) nor the Group x Time interaction (*p* = 0.920) were significant.

*Total level score:* The total score of children improved over time (*F* (2, 66) = 58.58, *p* < 0.001, η_p_^2^ = 0.64). Neither the Group factor (*p* = 0.288) nor the Group x Time interaction (*p* = 0.141) were significant.

*Overall communication domain* (Figure 1): There was a significant main effect of the factor Time (*F* (2, 66) = 64.67, *p* < 0.001, η_p_^2^ = 0.66) and of the Time x Group interaction (*F* (2, 66) = 5.24, *p* = 0.008, η_p_^2^ = 0.14). While the ESDM group improved from T0 to T2 (T0 vs. T1: mean difference = −39.89, *p_t_* < 0.001; T1 vs. T2: mean difference = −29.26, *p_t_* < 0.001), the TAU group did not (T0 vs. T1: mean difference = −20.56, *p_t_* = 0.050; T1 vs. T2: mean difference = −18.18, *p_t_* = 0.113).

*Overall socialization domain* (Figure 1): There was a significant main effect of the factor Time (*F* (2, 66) = 43.80, *p* < 0.001, η_p_^2^ = 0.57) and of the Time x Group interaction (*F*(2, 66) = 3.54, *p* = 0.035, η_p_^2^ = 0.10). The ESDM group improved from T0 to T2 (T0 vs. T1: mean difference = −31.32, *p_t_* < 0.001; T1 vs. T2: mean difference = −20.21, *p_t_* = 0.013), but the TAU group did not (T0 vs. T1: mean difference = −13.44, *p_t_* = 0.303; T1 vs. T2: mean difference = −16.37, *p_t_* = 0.124).

*Overall cognitive−play domain* (Figure 1): The score of children improved over time (*F* (2, 66) = 49.85, *p* < 0.001, η_p_^2^ = 0.60). Neither the Group factor (*p* = 0.117) nor the Group x Time interaction (*p* = 0.177) were significant.

*Overall motor domain* (Figure 1): The overall motor score improved over time (*F* (2, 66) = 55.11, *p* < 0.001, η_p_^2^ = 0.62). Neither the Group factor (*p* = 0.611) nor the Group x Time interaction (*p* = 0.361) were significant.

*Total score of the communication, socialization, cognitive−play, and motor domains* (Figure 1): There was a significant main effect of the factor Time (*F* (2, 66) = 53.27, *p* < 0.001, η_p_^2^ = 0.62) and of the Time x Group interaction *(F* (2, 66) = 4.49, *p* = 0.015, η_p_^2^ = 0.12). While the ESDM group improved from T0 to T2 (T0 vs. T1: mean difference = −110.58, *p_t_* < 0.001; T1 vs. T2: mean difference = −74.58, *p_t_* = 0.003), the TAU group did not (T0 vs. T1: mean difference = −47.06, *p_t_* = 0.222; T1 vs. T2: mean difference = −58.19, *p_t_* = 0.069).

#### 3.2.5. BOSCC

*Social communication*: Children of both groups improved their score over time (F (2, 66) = 17.66, *p* < 0.001, η_p_^2^ = 0.37), but neither the Group factor (*p* = 0.314) nor the Group x Time interaction (*p* = 0.270) were significant.

*Restricted and repetitive behaviors:* Neither the factors Time (*p* = 0.656), nor the Group (*p* = 0.723), nor the Group x Time interaction (*p* = 0.361) were significant.

*Total score*: There was a significant main effect of the factor Time (F (2, 60) = 11.67, *p* < 0.001, η_p_^2^ = 0.28), but not of the factor Group (*p* = 0.378) and the Group x Time interaction (*p* = 0.365).

## 4. Discussion

In this community-based study, we examined the feasibility and the effects of the ESDM delivered at low intensity (2 h per week) within the Italian Public Health System Services. Our aim was to extend previous ESDM implementation work in Italy [4,9,10] by administering the intervention to a larger number of children. We compared the experimental group to a non-randomized control group receiving TAU concurrently within settings similar to those in which the ESDM treatment was administered. Given the study interruption caused by the upsurge of COVID-19 in March 2020 in Italy, we herein reported results of a smaller number of children who completed the intervention and the follow-up assessments prior the beginning of the pandemic. Effectiveness was evaluated by comparing treatment outcomes of young children receiving the ESDM with those of children receiving TAU within a similar setting and with a similar intensity. We predicted that the ESDM group would make more progresses in cognitive, language adaptive, and social functioning in comparison to the control group. Feasibility was evaluated on the basis of procedures described in Vivanti et al. [5], Colombi et al. [9], and Holzinger et al. [22].

Over the last few years, several ESDM effectiveness and feasibility studies have been conducted in communities outside the USA [9,22,38,39,40]. These studies reported significant improvements in cognitive and language abilities, as well as in social skills. Moreover, feasibility seems well supported in such attempts as demonstrated by high participant retention rate both during intervention and at follow-up assessments, high level of satisfaction reported by families and professionals, fidelity of treatment, and promotion of the ESDM by state-funded programs. Compared to previous effectiveness studies conducted in Italy [4,9], our study partially supports their findings and overcomes possible methodological limitations both for the presence of a control group treated concurrently within comparable setting and for a greater homogeneity of the sample in terms of children age and numerosity.

The first aim of the current study was to evaluate the feasibility of the ESDM in the context of community-based Italian services that usually have limited resources. In line with other authors’ experience [9,22], we found that the ESDM approach is well accepted by parents because they appreciate high-quality teaching embedded within playful and naturalistic activities permeated by positive affect. Moreover, the multidisciplinary approach fits well with the organization of the Italian Public Health System, in which treatment is usually delivered by a team of professionals with a variety of expertise. In our study, the limitation of resources in terms of number of staff members and dedicated treatment space represented a challenge for therapists. On the other hand, they appreciated the emphasis on positive affect and nonverbal communication, perhaps because these components fit well with the Italian culture. The majority of both therapists and parents, however, stated that they felt that ESDM was a suitable approach for their young children with autism and that such approach was effective.

Our findings are partially consistent with previous ESDM studies conducted in university as well as community settings [18,41,42]. We found that children receiving the ESDM made more improvements in comparison to the TAU group in expressive and receptive language and social skills as measured by the ESDM Curriculum Checklist. Moreover, the ESDM group showed more improvement in the area of maladaptive behaviors as measured by the ESDM Curriculum Checklist in comparison to the control group. This observation finds support in several studies that have described [43,44] how problem behaviors (aggressive, disruptive, self-injurious) are significantly more common in children with autism than in other neurodevelopmental disorders, and how they are strongly correlated with the severity of receptive language difficulties and more severe socio-communicative deficits [45].

Additionally, at the end of the intervention, the ESDM group in comparison to the control group showed lower ASD symptoms in the social communication area as measured by the social affect (SA) comparison severity scores (CSS) of the ADOS-2. On the other hand, we did not observe any improvement in the area of restricted and repetitive behaviors (RRB) in either group. We believe this is consistent with what Estes et al. [46] observed in the follow-up study conducted two years after the conclusion of an early intervention with the ESDM, namely, that reducing the severity of core symptoms of autism requires further treatment and takes longer before it is observed. Perhaps, a similar explanation can be hypothesized for interpreting the finding that in our study both groups equally improved in adaptive functioning as measured by VABS-II, even though only the ESDM group showed a trend towards improvement in daily living skills over one year of treatment.

Finally, both treatment groups equally improved in cognitive and language abilities as measured by Bayley-III Scales and WPPSI-III, as well as adaptive functioning as measured by the VABS-II. Similar results were reported by Vivanti et al. [5], with the exception of receptive language, which increased significantly more in the ESDM group in comparison to the control group. It is possible that such differences may be related to the use of different instruments, given that Vivanti et al. [5] used the Mullens Scales of Development (MSEL) [47] to measure verbal and nonverbal developmental abilities. Additionally, such difference might be attributed to differences in treatment intensity given that in our study children received intervention for only 2 h per week over the course of 1 year. Our current findings confirm those reported in our pilot single group pre/post-treatment study [4], in which we observed improvement in cognitive and language development as measured by the same tools in a group of 21 young children with ASD who received the ESDM intervention in early development. In the study of Colombi et al., children participating in the ESDM group made more robust gains in language and cognition as measured by standardized assessments. Perhaps the difference found in our study may be attributed to the lower intensity of the intervention given that we delivered the ESDM for only two hours per week in comparison to 6 h per week delivered in Colombi et al. Similarly, it is possible that the lack of significant differences found in the BOSCC Scale were due to the low intensity delivery of the intervention. These results should be taken very seriously as it suggests that two hours per week may not be ideal to reach optimal outcomes. We certainly need to find ways to increase intensity. For example, we could implement parent-mediated ESDM intervention to increase productive learning time spent by the child throughout the day by teaching treatment strategies to the parents. Furthermore, in Italy we should try to find ways to establish better communication and collaboration between the school and the public health system. The Italian School System offers a special education teacher to each child with difficulties within an inclusive setting. Such special education could become high-quality learning time if the teachers were well trained in ASD-specialized intervention and worked closely with the public health system professionals.

It is worth pointing out that on the basis of the work of Contaldo et al. [10], we identified significant improvements by using the ESDM Curriculum Checklist [33], a measure seldom implemented to capture changes as effect of treatment. Through the ESDM Curriculum Checklist, we found continuous significant improvements over the course of the 12-month intervention in the ESDM group in the communication domain, in both the expressive and receptive subscales, and in the socialization domain, as well as in imitation abilities. On the other hand, the TAU group showed improvement in receptive language during the first six months of treatment.

The results of the current study indicate that the most sizeable gains achieved by the children treated with ESDM intervention are in developmental areas covered in the ESDM Curriculum Checklist, including receptive and expressive communication, social skills, and imitation ability, which are taught in the ESDM procedures with special emphasis. In particular, we found gain in imitation abilities that, as reported by Vivanti et al. [13], are considered predictive of ESDM treatment response. Although this finding has not been univocally observed in the literature [10,14], it is reasonable to hypothesize that since imitative skills are closely related to socio-communicative development, the ability to imitate also facilitates the co-construction of shared activities developed by therapist and the child, the ESDM framework of teaching.

Our study has limitations common to community effectiveness studies, firstly because it is not possible to maintain the level of experimental control that is typical of randomized control trials, such as randomization of participants, blindness of assessors, and stringent inclusion and exclusion criteria. On the other hand, the main focus of effectiveness studies is to evaluate the translation of evidence-based intervention programs, developed through rigorous university-based studies with a high level of resources, into community settings with much more limited resources. Additionally, the intervention administered to the control group was TAU, a non-manualized intervention without specific component nor fidelity tool. Thus, we do not have information regarding the quality, nor the treatment components implemented in the control condition. On the other hand, TAU is highly representative of interventions usually delivered in the Italian public health system. The impact of COVID-19 on the study can be considered a limitation, as it impacted the sample size because only part of the recruited children completed their follow-ups. However, we do not consider this to affect the interpretation of the results because the data reported in this study only concerned children who had completed their follow-up before the pandemic occurred.

We believe that our study has some strengths. Firstly, in our study, we compared the experimental group with a control group of children receiving intervention within similar community settings. Secondly, the high level of treatment fidelity achieved by ESDM prior the beginning of the intervention and maintained throughout the study seems to further support the feasibility of such a specialized treatment in the Italian public health system. Additionally, the parents appreciated many crucial aspects of the ESDM approach, including promoting the child’s spontaneity, as well as taking upon themselves a more active role in the treatment. Another strength is the implementation of outcome measure based on direct child’s observation including two innovative measures in research, which are the BOSCC and the ESDM Curriculum Checklist used as measures of change in response to treatment. On the basis of Reichow’s criteria [48] defining evidence-based practices (EBP) for children with ASDs, we believe that our study is of adequate quality. We think that the importance of our study, beyond the specific results regarding children’s outcomes, consists in showing that it is possible to deliver an evidence-based intervention, the ESDM, within the Italian public health system, a context in which usually non-ASD-specialized interventions are commonly delivered. Moreover, since it can be implemented in a variety of contexts and by a variety of professionals, the ESDM seems to fit Italian services well, wherein ASD intervention is delivered by a multidisciplinary team.

These findings must be interpreted cautiously of course, given the limitations inherent in the low intensity of treatment and small sample size. It is worth noting that intensity of intervention is still an open question in early intervention research, given that dosage studies are still in their infancy and that the number of treatment hours are not so clearly related to the treatment response [5,10]. However, the current study document that implementation of the ESDM in the Italian public health system, a non-English-speaking setting, can be feasible and effective. Our future directions include evaluating optimal treatment intensity and maintenance of intervention effects over time. Moreover, we hope to evaluate whether it is possible to implement the ESDM within a parent coaching and a school setting in order to increase treatment intensity by utilizing available and valuable resources.

## Figures and Tables

**Figure 1 brainsci-11-01191-f001:**
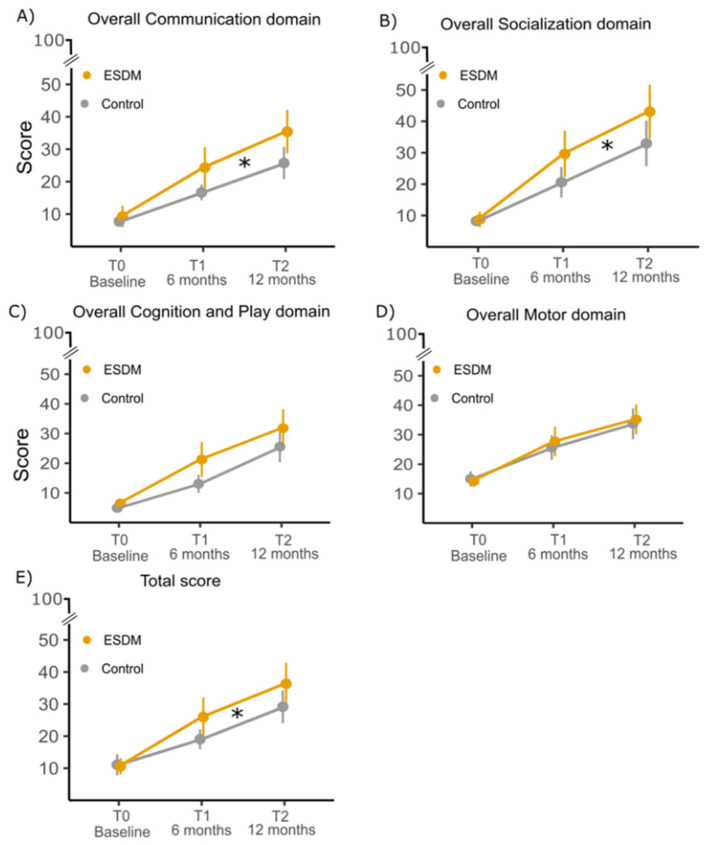
Group means and 95% confidence intervals expressed as percentage relative to ESDM checklist domain scores of the ESDM group (*n* = 19) and TAU group (*n* = 19). Scores in the communication (**A**); socialization (**B**); cognition and play (**C**); motor domain (**D**) and the combination of the communication, socialization, cognitive–play, and motor (**E**) areas of the ESDM group increased after treatment in comparison to the TAU group. Asterisks indicate significant differences between groups.

**Table 1 brainsci-11-01191-t001:** Demographic characteristics of the sample.

	ESDM (*n* = 19)	TAU (*n = 19*)
	**Mean (SD) or Number (%)**	
Children’s age at baseline (months)	29.6 (6.5)	30.5 (3.3)
Maternal age at conception (years)	31.8 (5.1)	32.6 (4.5)
Paternal age at conception (years)	34.9 (5.6)	35.2 (6.5)
**Sex**		
Male	17 (89.5%)	15 (78.9%)
Female	2 (10.5%)	4 (21.1%)
**Nationality**		
Italian	12 (63.2%)	13 (68.4%)
Others	7 (36.8%)	6 (31.6%)
**Prematurity (<36 weeks)**	1 (5.3%)	2 (10.5%)
**Siblings with ASD**	1 (5.3%)	2 (10.5%)
**Kindergarten attendance**	16 (84.2%)	14 (73.7%)
**Maternal educational level**		
Middle school	7 (36.8%)	6 (31.5%)
High school	5 (26.3%)	9 (47.4%)
University degree or higher	6 (31.5%)	4 (21 %)
Other/missing	1 (5.3%)	0 (0%)
**Paternal educational level**		
Middle school	7 (36.8%)	6 (31.5%)
High school	6 (31.5%)	12 (63.2%)
University degree or higher	5 (26.3%)	1 (5.3%)
Other/missing	1 (5.3%)	0 (0%)
**Maternal occupational status**		
Employed	12 (63.1%)	9 (47.4%)
Unemployed	7 (36.8%)	10 (52.6%)
Other/missing	0 (0%)	0 (0%)
**Paternal occupational status**		
Employed	17 (89.4%)	19 (100%)
Unemployed	1 (5.3%)	0 (0%)
Other/missing	1 (5.3%)	0 (0%)

SD: standard deviation.

**Table 2 brainsci-11-01191-t002:** Parent and therapist questionnaires to assess feasibility (1 = strongly disagree with the statement; 2 = slightly disagree with the statement; 3 = slightly agree with the statement; 4 = strongly agree with the statement.).

Parent Questionnaire	1	2	3	4
**Acceptability**				
I am convinced that ESDM is an appropriate treatment for my child				
I believe the therapy to be useful				
I felt sufficiently involved in the treatment by the therapists				
My child’s interests and preferences should be taken into consideration in order for the therapy to be successful				
It is important my child’s social interactions be engaging during the treatment				
It is important for the therapy to be based on play				
**Practicality**				
I managed to put therapy goals into practice in our daily routine				
I have had enough time to dedicate to my child in order to put the therapist’s teachings into practice				
Overall, undertaking the therapy has been very challenging				
**Efficacy**				
I believe my child has progressed more quickly thanks to ESDM				
My child has more fun playing with me now				
I play a more active role now when playing with my child				
**Therapist Questionnaire**	1	2	3	4
**Acceptability**				
I am convinced that ESDM is an appropriate therapeutic approach for young children				
I think ESDM therapy is useful				
There is no better therapy for young children than ESDM				
**Practicality**				
The aims and principles of the therapy can be easily integrated in my working practice				
Cooperation between parents and teachers works well				
Available areas and materials in my facility were adequate with regards to EDSM implementation				
**Efficacy**				
I believe the child progresses more quickly thanks to ESDM				

**Table 3 brainsci-11-01191-t003:** Summary of results for parent and therapist questionnaires.

	Acceptability	Practicality	Efficacy
	Mean	Mean	Mean
**Parent questionnaire**	3.71	3.88	3.31
**Therapist questionnaire**	3.70	2.85	3.66

## Data Availability

Data are available from the Open Science Framework: (DOI 10.17605/OSF.IO/9W3ZF): https://osf.io/9w3zf/ accessed on 15 February 2021.

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
