# Peer review of "Feasibility and Outcomes of the Early Start Denver Model Delivered within the Public Health System of the Friuli Venezia Giulia Italian Region"

_brainsci, 2021, doi:10.3390/brainsci11091191_

Round 1

Reviewer 1 Report

First off, this is a well-written paper that seeks to do what too few scientists do- which is to do studies in different contexts/populations to see if the research is generalizable or not. Thank you for researching this!

I would like more details about TAU (Lines 295-300 on page 9). What sort of services are representative of those usually delivered in Italy? What are some examples of non-specialized ASD interventions? As someone not from Italy, that is important for me to know to evaluate whether TAU is a reasonable control for EDSM in all contexts or just the Italian context.

Also, over a 12 month period, what is the expected trajectory for the acquisition of the skills taught in ESDM for children without an ASD diagnosis? That would be a useful metric to compare the progress of both the ESDM and TAU groups. Or has this been done with other populations (e.g. ID)? 

On page 15, lines 469-474, I am confused about how there is 100% retention when you list people who dropped out of the study at the beginning (page 3). Maybe I am misreading it? If so, try to clarify for people like me who may have misinterpreted what you are saying.

On page 16, line 538, the grammar is off here "We suppose that consistently with what Estes . . .". I think you mean "We believe this is consistent with what Estes . . ."

On page 17, line 583, I think the grammar is off here "We can state that the most sizeable gains achieved . . . ". I would replace "We found that the" or "The results of the current study indicate that the . . . ".

Like most research in the time of COVID, this study was dramatically impacted both in timeline measurements and in overall delivery time and setting. You mention that earlier in the paper, but it really should be spelled out again in the limitations section. Why and how did COVID impact your study and what sort of caution should we take when we interpret your results? 

Overall, this is an excellent paper that contributes to the field in important ways. I recommend it be accepted after minor revisions.

Reviewer 2 Report

While these authors are to applauded for attempting to report on their study during the pandemic, unfortunately, an alternative to the approach taken may have been better for making a novel and helpful contribution to the field, especially given the very limited literature that exists thus far on effectiveness studies.

Since there are very few actual effectiveness studies in ASD at all, it is important for such a study to be statistically powered to produce meaningful findings, and this study was clearly not, given the very small sample sizes that ended up being reported due to pandemic-related delays in data collection, creating missing data (that really can’t be captured even now, since time since treatment is critical here).  So with a sample size a fraction of the original attempted in this study, the authors continued to try to match ESDM treatment group to a treatment as usual, based on age and sex.  However, the authors then went on an conducted MANY analyses (too many, in my opinion, even if the entire sample had been collected), using multiple types of statistics (claiming to add a Bayesian analysis due to the small sample size, but this just seemed to cause multiple analyses to be done, which does not seem most parsimonious?).    It also seems unhelpful that much data, from the remaining TAU group that was not matched, is just being thrown away. At the least, these data (the full TAU sample) should be shown in a supplemental table, but given my recommendations below, it may be considered for the primary text.

Beyond these concerns, my biggest concern about this study was its use of a secondary aim to capture “feasibility” of the intervention. This did not make sense to me; frankly, it did not make sense to see feasibility in an effectiveness study at all, but if it is to be there, seems like it would make more sense for feasibility to be aim 1, and only if that aim is met (a threshold of feasibility is predefined and then criteria are met) would the authors move on to conducting analyses of actual effectiveness.  But this threshold for feasibility should be set a priori. So given the sample size limitations, I think it would have been wisest, and most prudent to ONLY report on feasibility in this sample.

If the authors feel like they need to report on effectiveness, it is recommended that they greatly scale back analyses.  While the authors say on pg 3 that they are basing on previous research that the ESDM group would demonstrate higher gains in language, cognitive, social and daily living – this can get much more specific based on specific previous ESDM findings. The study should be powered on JUST ONE primary outcome, and selective secondary outcomes.  If/when analyses are conducted/reported, they don’t need to be spelled out in as much detail as they are here – there is some redundancy between tables and text.  But there are too many here so reduction seems necessary.

It is suggested that the authors look at this paper: https://www.ncbi.nlm.nih.gov/books/NBK44029/pdf/Bookshelf_NBK44029.pdf and others to determine how an effectiveness study is supposed to differ from an efficacy study, including a larger, more generalizable sample size, if possible (so less stringent study criteria) and use a realistic time frame (is 1 year realistic for generalizable change to be seen in young children with ASD?).  Most effectiveness studies are randomized (and blinded, although we know this is difficult with ESDM), with allocation of group concealed – much trial terminology like this was not used in the current study, and it seems essential to add the open label nature of this study as well.  I do not think what is presented here truly represents an effectiveness study (and I think this is the case even if the study had not been interrupted by the pandemic).

Regarding analyses, if the authors choose to try to publish the limited data from the study measures, I think no statistics would make more sense here, perhaps just showing the growth.  However, if/when statistics are done, why were regression analyses considered?  

Reviewer 3 Report

This article presents a quasi-experimental study to determine the feasibility and results of the ESDM program provided by a public health service in Italy (Friuli Venezia Giulia region).
The study is well designed and well developed although its power has been reduced as a consequence of the COVID-19 pandemic by reducing the sample. The measures taken are adequate.

Here I present some observations that I believe can improve the work:

Regarding the introduction, I believe that the results of the meta-analysis developed by Fuller et al (2020) and published in this journal (https://doi.org/10.3390/brainsci10060368) should be introduced. Although the ESDM is well known, I think it is timely to acknowledge the evidence of the results through a meta-analysis.
On line 156 (Materials and Method section, design subsection) it is stated that the diagnostic evaluation protocol was administered at the initial time...." If it refers to the ADOS-2 it would be clearer if this instrument were specified and if it refers to the set of measures (ADOS, Vineland, Baley, WPSSI, etc) it should be specified at some point that this (the set of instruments) is the diagnostic assessment protocol.

In the section describing the measures (lines 197 and following) the instruments are cited erroneously, I believe. For example, the original version of the Baley III is correctly cited (citation number 27) and the Italian version (supposedly) citation 26. Here I believe that the authorship of the Baley III is attributed to Ferri et al 2006 when it should be only the translation and adaptation of the same. It can be solved by quoting the original and the corresponding Italian version in the same citation. It should be revised in all the instruments described.

It is striking that WPPSI-III is used when a more current version already exists. It could be justified by the non-existence of the Italian version of version IV.

The ESDM curriculum checklist is presented in lines 225 and following lines. This checklist can be answered by the therapist, the parents or the educators but it is not reported who answered it and if it was answered by several of them, what were the consistency indices between judges?

Author Response

Dear Reviewer,

Thank you for your comments and suggestions. You will find the implemented revisions in the main manuscript. Below, our replies to your concerns.

As you suggested, we have included in the Introduction section the summary of the results of the meta-analysis study by Fuller et al. (2020). You can find it at line 87.

With reference to the diagnostic protocol, we have now specified that the diagnostic evaluation protocol includes the whole set of measures (ADOS-2, VABS-II, Bayley-III, WPSSI-III, BOSCC, ESDM Curriculum Checklist). See line 162

Concerning the quotations of the measures, we have now sorted the references according to your suggestions (original work first, then the translated version). 

We used the WPPSI-III version because WPPSI-IV was validated and translated in 2019, after the beginning of the recruitment for this study. We specified this note at line 213.

The ESDM Curriculum Checklist was completed only by the therapists who treated the child. We have added this specification to line 247.

Kind regards,

Raffaella Devescovi, MD